# Ultrasound Muscle Assessment and Nutritional Status in Institutionalized Older Adults: A Pilot Study

**DOI:** 10.3390/nu11061247

**Published:** 2019-05-31

**Authors:** Alvaro Mateos-Angulo, Alejandro Galán-Mercant, Antonio Ignacio Cuesta-Vargas

**Affiliations:** 1Department of Physiotherapy, IBIMA, University of Málaga, 29017 Málaga, Spain; amateos@uma.es (A.M.-A.); acuesta@uma.es (A.I.C.-V.); 2MOVE-IT Research group and Department of Nursing and Physiotherapy, Faculty of Nursing and Physiotherapy University of Cádiz, 11002 Cádiz, Spain; 3Biomedical Research and Innovation Institute of Cádiz (INiBICA) Research Unit, Puerta del Mar University Hospital University of Cádiz, 11002 Cádiz, Spain; 4School of Clinical Sciences of the Faculty of Health at the Queensland, University of Technology, 4072 Brisbane, Australia

**Keywords:** ultrasound, muscle thickness, functional capacity, nutritional assessment, aging

## Abstract

Muscle thickness, measured by ultrasonography, has been investigated for nutritional assessment in older adults, however the associations between muscle ultrasound parameters in the lower limb and nutritional status have not been studied. The aim of this study was to investigate the relationship between muscle thickness echo intensity (EI), and nutritional status in home care residents. A cross sectional study was conducted involving 19 older adults from a home care in Malaga (Spain). We evaluated lower leg muscles by ultrasound, anthropometric data, physical function (measured by gait speed and the Short Physical Performance Battery), strength (handgrip and knee extensors strength) and nutritional status across the Mini-Nutritional Assessment Short-Form (MNA-SF). We found that muscle thickness assessed by ultrasonography independently predicts nutritional status by MNA-SF and after adjusting for handgrip strength or age and sex. As secondary findings, we found relations between strength, functional capacity and the MNA-SF test. These results suggest that lower leg muscle ultrasound parameters could be used as a low-cost objective method for muscle evaluation in nutritional assessment in older adults.

## 1. Introduction

Malnutrition in elderly people increases morbidity and mortality, reduces quality life and increases length of hospital stay [1,2,3,4,5]. Malnutrition involves a reduction of muscle mass that can impact muscle strength [2], and it is related to functional impairment in nursing home residents [6]. In addition, malnutrition, or the risk of malnutrition, increases the risk of frailty in institutionalized older adults [7]. Therefore, a periodic nutritional evaluation of in home care residents is necessary to prevent morbidity and mortality [8].

Functional capacity is a very important health indicator in older adults and is related to frailty and cognitive status in nursing home residents [9]. In addition, the relationships between functional and nutritional status have been reported previously [10]. Short Physical Performance Battery (SPPB) and Gait Speed are the most recommended tests to evaluate physical function in research and clinical practice [11,12].

Ultrasound is an ambulatory and low-cost tool that is valid and reliable for assessing specific muscles [13]. Previous studies have investigated the adductor pollicis muscle thickness using ultrasonography for nutritional assessment [14,15]. However, lower limb muscles are the most important muscle groups for functional independence in older adults [16]. Muscle thickness and echo intensity ultrasound parameters in lower limb muscles have been associated with functional capacity in older adults [17,18,19]. For this reason, it is important to investigate the interactions between lower limb muscle ultrasound parameters and nutritional status in older adults.

In the elderly population, malnutrition is an important problem because it involves an increase in frailty risk, reduces quality of life and increases mortality. For this reason it is necessary to explore new tools that can identify better nutritional profiles. The aim of this study was to investigate the associations between muscle ultrasonographic quantitative and qualitative assessment, the physical and functional parameters related to frailty and the risk of malnutrition using the Mini-Nutritional Assessment (MNA) in institutionalized older adults. We studied muscle thickness and echo intensity by ultrasound in lower leg muscles, handgrip strength (HGS) and knee extensor strength, gait speed, the SPPB test, anthropometric parameters and the MNA test. A secondary aim was to study the reliability of ultrasonographic muscle thickness measurements of the gastrocnemius medial and tibialis anterior muscles.

## 2. Materials and Methods 

### 2.1. Study Subjects

A total of 19 institutionalized older adults (14 women and 5 men), aged 70–100 years, were recruited between February and June 2016 from a geriatric center in Málaga (Spain). Each participant received a detailed explanation of the study and gave written informed consent before participation. The study complied with the principles laid out in the Declaration of Helsinki and was approved by the ethics committee of the Faculty of Health Sciences at the University of Malaga, Spain. Inclusion criteria were: (1) males and females, aged 70–100 years old; (2) body mass index (BMI) between 18.5 and 30; (3) the ability to speak and understand Spanish. Exclusion criteria were individuals with recent surgery, fractures or any condition or acute processes that does not allow them to complete the study or that could affect the results of the evaluations.

### 2.2. Study Design

The study used a cross-sectional research design that examined the relationship between anthropometrics parameters, muscle strength, muscle architecture by ultrasonography and nutritional status measured by the MNA in institutionalized older adults. Men and women between 70 and 100 years were recruited and evaluated in a geriatric center. The subjects signed a written informed consent form and completed the evaluations in a single day session.

### 2.3. Procedures

#### 2.3.1. Anthropometric Measurements

The height and weight were recorded with the participant barefoot and in light clothing. The participant, standing in the anatomical position with the occipital region, back, gluteal region and heels in contact with the height rod, took a deep breath for height measurement. The height is the distance from the vertex to the soles of the feet. Calf circumference on the dominant side was measured at the point of the widest diameter of the calf. Mid-upper arm circumference on the dominant side was measured on the upper left arm, flexed at 90°, at the midpoint between the acromion and the olecranon. The anthropometric procedures described were obtained following the guidelines of The International Society for the Advancement of Kinanthropometry (ISAK) [20].

#### 2.3.2. Nutritional Assessment

Nutritional status was evaluated by the MNA Short-Form (MNA-SF) test. MNA-SF is a validated screening tool to help identify elderly people who are malnourished or at risk of malnutrition [21]. MNA-SF consists of six questions, including the decline of food intake over the last three months, weight loss during the last three months, mobility, stress or acute disease in the last three months, neuropsychological problems and BMI or calf circumference. From these questions the test score was calculated between 0 and 14 points. If the score was 12–14, patients were considered to have a normal nutritional status, scores of 8–11 indicated a risk of malnutrition and scores under 7 were considered malnourished.

#### 2.3.3. Muscle Strength and Physical Function

Bilateral handgrip and bilateral knee extensor strength (KES) were measured. The HGS was calculated using a hydraulic hand-held dynamometer (Saehan Corporation, Masan Free Trade Zone, Korea) [22]. HGS was assessed following the recommendations of the American Society of Hand Therapists (ASHT) [23]. During the measurements, the subjects sat with their feet touching the ground, their elbows flexed to 90° and their forearms in the neutral position. All participants performed three trials with one-minute rest, alternating between their dominant and non-dominant hand. The highest value from each side was recorded as maximal grip strength.

A digital manual dynamometer POWERTRACK^®^ JtechMedical was used to evaluate maximum knee extensor muscle strength. The participant was placed in a sitting position on a stretcher with his hands resting on his legs and feet hanging off the ground. The examiner placed one hand to stabilize the subject’s leg and the other hand to support the load cell on the subject’s distal third tibia. Starting from 90° knee flexion, the subject performs a knee extension resisted by the examiner with the dynamometer’s load cell. A full extension was avoided, with the knee flexion reaching up to 5°. The maximum peak force was recorded from the dynamometer´s digital display in Newtons. The test was performed three times for each subject, with a 2-minute break between tests; the highest value was recorded [24].

To assess functional capacity, the SPPB test was used [25]. SPPB consists in three objective physical tests: balance, 4 meter gait speed and sit-to-stand ability. Every test scored from 0 to 4, resulting in 0 to 12 for the SPPB score. In addition, the four-meter gait speed test, scored in meters per second (m/s), was also recorded as an independent variable.

#### 2.3.4. Ultrasonography

The following lower limb muscles of the dominant side were examined using ultrasound B-mode images (Figure 1): medial gastrocnemius (MG) and tibialis anterior (TA). A longitudinal static B-mode image of each muscle on the dominant side of the patient were acquired using an ultrasound device (Esaote MyLab One) with a linear array transducer (SL3323) with a variable frequency up to 22 MHz. The same evaluator carried out a second ultrasonography of the MG and TA on the participants to examine the test–retest reliability of the measurements. A large amount of ultrasound gel was used in order to minimize the pressure applied on the skin and to increase the quality of the images. 

Representation of superficial and deep aponeuroses was optimized in order to achieve the best muscle image possible. The transducer was placed at the anatomical location corresponding to the largest diameter of the muscle examined, described in a previous study; the MG from the mid-sagittal line of the muscle, the midway between the proximal and distal tendon insertions and the TA at one-quarter of the distance from the inferior aspect of the patella to the lateral malleolus were examined.

Muscle images were analyzed offline with ImageJ software. To calculate MT, a known distance of 1 cm, as shown in the image, was used to calibrate the software program. The two images were measured and averaged. Muscle thickness was calculated as the distance between the superficial and deep aponeuroses. Lower Leg MT was calculated as the average of GM and TA MT. A region of interest was selected including as much as possible of the muscle area of the muscle but excluding the muscle fascia, using polygon selections for the EI calculation. EI was determined by grey-scale analysis function and expressed in arbitrary units as a value between 0 (black) and 255 (white) [26,27].

### 2.4. Statistical Analysis

A database was created with the information gathered from the experimental session in order to analyze the results. The Shapiro–Wilk test was used, as determined by the normality of distribution variables. Interclass correlation coefficient (ICC 3,1) was used to determine intrasession and inter-rater reliability of the ultrasound muscle images. Descriptive statistics were performed with measures of central tendency and dispersion of variables. Inferential statics were performed with bivariate correlations with the Pearson and Spearman correlation test and linear regression analysis. All statistical analyses were performed with SPSS 21 (SPSS Inc., Chicago, IL, USA) software package.

## 3. Results

A second ultrasonography was carried out in all the participants to examine the intraclass correlation coefficient of the MG and TA muscle thickness measurement in repose and maximum voluntary contraction (MVC). The ICC value for the TA muscle thickness measurement was 0.95 in repose and 0.98 in MVC. The ICC value for MG muscle thickness measurement was 0.97 in repose and 0.99 in MVC. Table 1 shows the participant´s physical characteristics and MT and EI values.

Lower leg MT was associated with calf circumference and lower leg EI was associated with arm circumference (see Table 2). Significant correlation coefficients were found between muscle thickness and MNA-SF, but not between echo-intensity and MNA-SF (see Table 3). In addition, significant correlations were found between the MNA-SF test and HGS, GS and SPPB, but not between MNA-SF and KES (see Table 4).

Table 5 shows a summary of the best stepwise multiple regression analysis. GM muscle thickness in MVC independently explains about 44% of the variance of MNA-SF, and about 64% after adjusting by HGS. TA muscle thickness in MVC explains about 33% of the variance of MNA-SF, and about 63% after adjusting by age and sex.

## 4. Discussion

The present study evaluated the association between muscle ultrasound variables, strength, functional components of frailty and nutritional status on a population of institutionalized older people in Malaga (Spain). Muscle thickness and echo-intensity was associated with nutritional parameters. We found that muscle thickness assessed by ultrasonography independently predicts nutritional status by MNA, and after adjusting by HGS or age and sex. In addition, this study showed an excellent reliability of the MT measurement by ultrasonography. As secondary findings, we found relations between physical parameters (measured by HGS and KES), functional parameters (measured by Gait Speed and SPPB) and the MNA test. 

As far we know, this study is the first one to evaluate the association between lower limb muscle thickness and nutritional status by MNA. Ultrasound is a method to assess muscle mass, which has many advantages: it is a low cost tool, it can assess specific muscles and it is valid and reliable [13]. MT is a parameter that can be used to predict fat-free mass in the elderly population [28]. There are some reports that investigated the relationship between adductor pollicis MT and nutritional status [14,15]. Relations between muscle thickness and MNA in hospitalized older adults has been studied [29]. A previous study showed relations between muscle thickness and nutritional parameters (arm circumference) in institutionalized older adults [30]. 

Volpini et al. found significant correlations between adductor pollicis MT and arm circumference (*p* = 0.033) [30]. In addition, Hasegawa et al. investigated the relationship between temporal muscle and nutritional parameters and found significant correlations between temporal muscle thickness and arm circumference (*r* = 0.462, *p* < 0.001) and calf circumference (*r* = 0.608, *p* < 0.001) [31]. In this way, the present study showed significant correlations between lower leg MT and calf circumference (*r* = 0.545, *p* = 0.016), but not arm circumference. On the other hand, arm circumference is a parameter that can be used to detect undernutrition in elderly people [17]. The present study showed an association between lower leg EI and arm circumference (*r* = −0.478, *p* = 0.038). Greater muscle quality (lower EI values) was related with higher arm circumference, this finding suggests that EI could be used as a nutritional parameter. No previous studies have investigated the relationship between EI and nutritional parameters.

This is the first study that has associated muscle ultrasound parameters in the lower limb with the nutritional status measured by MNA. Schwanke et al. showed associations between MT and MNA, but the muscle studied was the adductor pollicis [29]. In this line, we found in the present study that multiple regression models associated lower leg MT with MNA scores, even after adjusting by strength, age or sex. This finding suggests that lower leg MT can be an indicator of nutritional status in elderly people. Therefore, lower leg MT can be applicable in clinical practice for nutritional geriatric assessment. The advantage of measuring lower limb muscles in the geriatric population is that lower limb muscles are the most important muscles for functional independence [16]. In addition, lower leg MT is related to functional capacity and strength [32]. Therefore, it seems to be more interesting to assess lower leg muscles by ultrasonography in elderly people because, in addition to nutritional information, lower leg MT is an interesting parameter for general geriatric evaluation. By measuring muscle ultrasound parameters in only one part of the body, we can obtain an indicator of functional capacity and nutritional status. For this reason, lower leg MT evaluation has a high applicability in clinical practice.

Functional status measured by Gait Speed and SPPB tests are related to frailty [9] and cognitive function [33]. In addition, Gait Speed is considered to be the diagnostic test to identify frailty [34] and sarcopenia [12]. In this context, we found a significant association between MNA and functional capacity measured by Gait Speed (*r* = 0.502, *p* = 0.04) and SPPB (*r* = 0.597, *p* = 0.007). Similar to our investigation, previous studies found associations between MNA and functional status in older people [35,36]. Vivanti et al. found associations between nutritional status and functional capacity measured by the timed-up-and-go test [35]. In addition, Schrader et al. found that nutritional status according to MNA was related to functional status measured by the Barthel Index and timed-up-and-go test [36]. However, SPPB and Gait Speed tests used in the present study are the most recommended tests by international working groups for functional capacity assessment [11,12]. On the other hand, it is well known that HGS is a health status predictor, and in addition, it has been considered as a nutritional indicator [37]. In the present study, we found strong associations between HGS and MNA (*r* = 0.748, *p* = 0.0002).

The main limitation of the present study is the sample size. Further research is required to confirm the potential of lower limb muscle thickness to assess nutritional status in institutionalized older adults and to establish references standards, stratified by age and gender. The total number of ultrasound measurements in each subject, the methodological difficulties, and the off-line image processing hamper the achievement of a higher sample size. However, facilitating the access of the ultrasound measurements into the clinical practice has the advantages of being a simple, effective and non-invasive method to study the nutritional status of older adults. The very precise information provided by ultrasound, in combination with other parameters, makes the results from this sample size of great interest. Future studies replicating this approach in larger samples are required. On the other hand, another limitation of the present study is the lack of confirmative evidence with other malnutrition indicators. Further studies having confirmative sampling of malnutrition could be useful to improve the validity of the ultrasound measurement for identifying nutritional profiles in older adults.

In conclusion, a greater leg muscle thickness was independently associated with less risk of malnutrition in institutionalized older adults. Age, sex and strength also influenced nutritional status. Moreover, better muscle quality (lower echo-intensity) was associated with greater nutritional parameters (arm circumference). These results suggest that lower leg muscle ultrasound parameters could be used as a low-cost objective method for muscle evaluation in nutritional assessment in older adults.

## Figures and Tables

**Figure 1 nutrients-11-01247-f001:**
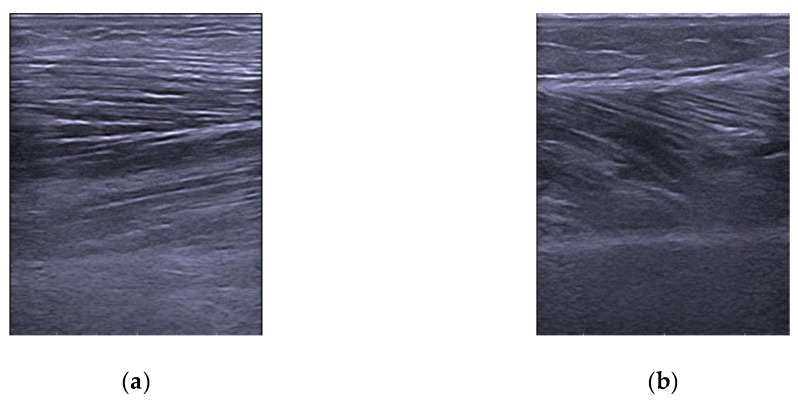
(**a**) Tibialis Anterior transverse ultrasound image, (**b**) Medial Gastrocnemius transverse ultrasound image.

**Table 1 nutrients-11-01247-t001:** Physical, nutritional and muscle ultrasound characteristics.

Variable	Mean (SD)
Age (years)	85.11 (7.12)
Height (m)	1.60 (0.05)
Weight (kg)	64.5 (9.43)
BMI (kg/m^2^)	25.22 (3.39)
Arm CC (cm)	27.87 (2.89)
Calf CC (cm)	33.85 (2.7)
MNA-SF (0-14)	11 (1.63)
HGS (N)	16.95 (4.42)
KES (N)	138.17 (63.14)
TA MT (cm)	2.28 (0.28)
TA MT MVC (cm)	2.49 (0.32)
GM MT (cm)	1.17 (0.33)
GM MT MVC (cm)	1.23 (0.34)
TA EI (0-255)	87.64 (7.57)
GM EI (0-255)	68.25 (11.05)

BMI: body mass index, CC: circumference, MNA-SF: Mini Nutritional Assessment Short Form, HGS: handgrip strength, KES: knee extensors strength, GM: gastrocnemius medial, MT: muscle thickness, MVC: maximum voluntary contraction, EI: echo-intensity, TA: tibialis anterior.

**Table 2 nutrients-11-01247-t002:** Correlations between muscle ultrasound variables and nutritional parameters.

	Calf Circumference		Arm Circumference	
Variable	Coefficient of Correlation	*p*-Value	Coefficient of Correlation	*p*-Value
Lower Leg MT	0.545	0.016	0.270	0.264
Lower Leg EI	−0.200	0.412	−0.478	0.038

EI: echo-intensity, TA: tibialis anterior.

**Table 3 nutrients-11-01247-t003:** Correlations between muscle ultrasound variables and the MNA-SF test.

Variable	Coefficient of Correlation	*p*-Value
GM MT	0.574	0.010
GM MT MVC	0.653	0.002
GM EI	−0.115	0.640
TA MT	0.710	0.001
TA MVC	0.595	0.007
TA EI	−0.208	0.394
Lower Leg MT	0.712	0.001
Lower Leg EI	−0.309	0.389

GM: gastrocnemius medial, MT: muscle thickness, MVC: maximum voluntary contraction, EI: echo-intensity, TA: tibialis anterior.

**Table 4 nutrients-11-01247-t004:** Correlations between physical function variables and the MNA-SF test.

Variable	Coefficient of Correlation	*p*-Value
HGS	0.748	0.0002
KES	0.439	0.060
GS	0.502	0.04
SPPB	0.597	0.007

HGS: hand grip strength, KES: knee extensors strength, GS: gait speed, SPPB: Short Physical Performance Battery.

**Table 5 nutrients-11-01247-t005:** Multiple Regression Analysis.

Dependent Variable	Predictor Variables	Standardized β	Adjusted R2
MNA-SF			
Step 1	Lower Leg MT	0.711 (*p* = 0.001)	0.506 (*p* = 0.001)
Step 2	Lower Leg MT	0.560 (*p* = 0.003)	0.638 (*p* = 0.0003)
	HGS	0.394 (*p* = 0.028)	
MNA-SF			
Step 1	GM MT MVC	0.665 (*p* = 0.002)	0.442 (*p* = 0.002)
Step 2	GM MT MVC	0.541 (*p* = 0.003)	0.643 (*p* = 0.0003)
	HGS	0.465 (*p* = 0.009)	
MNA-SF			
Step 1	TA MVC	0.573 (*p* = 0.010)	0.328 (*p* = 0.010)
Step 2	TA MVC	0.809 (*p* = 0.001)	0.534 (*p* = 0.002)
	Age	0.511 (*p* = 0.017)	
Step 3	TA MVC	0.624 (*p* = 0.002)	0.630 (*p* = 0.002)
	Age	0.322 (*p* = 0.067)	
	Sex	0.349 (*p* = 0.052)	

HGS: hand grip strength, GM: gastrocnemius medial, MT: muscle thickness, MVC: maximum voluntary contraction.

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
