# Peer review of "Ultrasound Muscle Assessment and Nutritional Status in Institutionalized Older Adults: A Pilot Study"

_nutrients, 2019, doi:10.3390/nu11061247_

Round 1
Reviewer 1 Report
Ultrasonography is a reliable method of measuring body fat and lean muscle status. The present study is interesting and will be of interest to the readers of the journal. Overall, this is a clear manuscript. The present study suggests that lower limp muscle thickness using ultrasonography can be an indicator of nutritional status.
My comments are as follows:
Given the relatively low number of participants, I recommend the title to me rephrased: Ultrasound muscle assessment and nutritional status in institutionalized older adults: a pilot study
Specific points
The aim of this study was to investigate the relationship between muscle thickness and nutritional status. I propose to add a comment about 1) what nutrition assessment is and 2) a brief comment about the problems that exist in relation to malnutrition and the elderly population.
Line 143. BMI, CC and MNA SF should be added in the description- below table
Line 151. GM, MVC and TA should be deleted
Line 171. The authors concluded the study showed an excellent reliability of the MT measurements by ultrasonography (Line 171-172) Can the authors comment how they performed reliability (methods section) and better explain results (Results section) ?
Lines 178- 180-181. There are too many repetitions (Previous study.. A previous study..). Rephrase.
Line 209. Indeed Gait speed is considered a diagnostic test to identify frailty. I recommend to add sarcopenia also (plus reference).
Line 273. The name of the group should be added: Working Group on Functional Outcome Measures for Clinical Trials. Functional outcomes for clinical trials in frail older persons: time to be moving. …
Author Response
#Reviewer 1#
Ultrasonography is a reliable method of measuring body fat and lean muscle status. The present study is interesting and will be of interest to the readers of the journal. Overall, this is a clear manuscript. The present study suggests that lower limp muscle thickness using ultrasonography can be an indicator of nutritional status.
My comments are as follows:
Given the relatively low number of participants, I recommend the title to me rephrased: Ultrasound muscle assessment and nutritional status in institutionalized older adults: a pilot study
Response 1: Thank you very much for your recommendation. The title has been modified.
Specific points
The aim of this study was to investigate the relationship between muscle thickness and nutritional status. I propose to add a comment about 1) what nutrition assessment is and 2) a brief comment about the problems that exist in relation to malnutrition and the elderly population.
Response 2: Thank you very much at this point. The introduction section has been modified to add this comments.
Line 143. BMI, CC and MNA SF should be added in the description-below table
Response 3: Thank you very much at this point. Table 1 description has been modified.
Line 151. GM, MVC and TA should be deleted
Response 4: Thank you very much at this point. Table 2 description has been modified.
Line 171. The authors concluded the study showed an excellent reliability of the MT measurements by ultrasonography (Line 171-172) Can the authors comment how they performed reliability (methods section) and better explain results (Results section) ?
Response 5: Thank you very much at this point. Methods section and Results sections has been modified to a better explanation of the muscle thickness reliability evaluation. In addition, the introduction section has been modified to add the reliability of muscle thickness measurements as a secondary aim.
Lines 178- 180-181. There are too many repetitions (Previous study.. A previous study..). Rephrase.
Response 6: Thank you for your suggestion. The discussion section has been modified to delete the repetitions.
Line 209. Indeed Gait speed is considered a diagnostic test to identify frailty. I recommend to add sarcopenia also (plus reference).
Response 7: Thank you very much for your suggestion. The discussion section has been modified to add your suggestion.
Line 273. The name of the group should be added: Working Group on Functional Outcome Measures for Clinical Trials. Functional outcomes for clinical trials in frail older persons: time to be moving. …
Response 8: Thank you very much. The reference list has been modified.
Reviewer 2 Report
the described topic is exciting and finding a reliable and time-saving tool to identify malnourished people is a crucial point to deal with this issue, especially in acute settings.
Authors decide to compare the ultrasound assessment with the mini nutritional assessment; that's one of the most used risk assessment scales worldwide. They don't specify if they had confirmative evidence with other malnutrition indicators, i.e. biochemical or others. To have confirmative sampling should be useful to confirm the assessment of malnourished and not the level of risk uprising from the MNA.
Do Authors consider if the reported BMI have been related to the health status of patients? In acute settings, patients should be involved in many complications that modify water retention or modify body composition.
In my opinion, Authors should define if they take note of this informations or describe those limits in their paper.
Author Response
#Reviewer 2#
The described topic is exciting and finding a reliable and time-saving tool to identify malnourished people is a crucial point to deal with this issue, especially in acute settings.
Authors decide to compare the ultrasound assessment with the mini nutritional assessment; that's one of the most used risk assessment scales worldwide. They don't specify if they had confirmative evidence with other malnutrition indicators, i.e. biochemical or others. To have confirmative sampling should be useful to confirm the assessment of malnourished and not the level of risk uprising from the MNA.
Response 1: Thank you very much at this point. We assume the limitation of our study of the lack of confirmative evidence with other malnutrition indicators. The discussion section has been modified to indicate this limitation.
Do Authors consider if the reported BMI have been related to the health status of patients? In acute settings, patients should be involved in many complications that modify water retention or modify body composition.
Response 2: Thank you very much at this point. Patients with acute processes has not been included in our study. The exclusion criteria have not been described properly in our manuscript. The methods section has been modified for a better explanation of the exclusion criteria.
In my opinion, Authors should define if they take note of this informations or describe those limits in their paper.
Response 3: Thank you very much for your suggestion. The methods section has been modified specifying better the exclusion criteria of the study. In addition, the discussion section has been modified describing the lack of confirmatory evidence with other nutritional indicators as a limitation of our study.
Round 2
Reviewer 2 Report
The paper has been significantly improved in its readability. this method needs a confirmative study to assess its reliability and effectiveness intercepting malnutrition in acute setting patients
Authors fulfilled all the suggestion required; the paper should be accepted this way